# Case Study of Underground Shield Tunnels in Interchange Piles Foundation Underpinning Construction

**Chengran Zhang, Yujia Zhao, Zhen Zhang * and Bing Zhu**

College of Civil Engineering, Southwest Jiaotong University, Chengdu 610031, China;
zhangchengran@my.swjtu.edu.cn (C.Z.); zyj10239@126.com (Y.Z.); zhubing126@126.com (B.Z.)
* Correspondence: zhangz@my.swjtu.edu.cn; Tel.: +86-17317051657

**Abstract:** Technology of shield tunneling has been widely used in modern city subway system building, however, due to the limit of urban construction land resources, it is inevitable for the new building structures conflicted with existing one. Thus, it is essential to taking correct measures to guarantee the safety of existing structure. In this case, a typical case of metro shield tunnel crossing through city overpass bridge piles underpinning construction project is studied by the numerical method and site field monitoring. According to the existing overpass bridge structure, geological conditions and site operation environment, a suitable construction method of piles foundation underpinning plan for this project is adopted. In order to ensure the structure safety of the existed overpass bridge during the pile foundation underpinning construction, a numerical simulation model is established which takes the whole foundation underpinning structure and construction steps into consideration. The numerical simulation result shows that the stress and the settlement of the piles foundation underpinning structure is reasonable. After the comparing of the site monitoring data with the numerical model simulation results, it is found that the site settlement and stress results are highly consistent with the calculated results, and it proves the feasibility of the piles foundation underpinning construction scheme. Eventually, the site gauging settlement data of the overpass pier and the pile cap indicate that the original overpass structure is barely affected by the shield tunnel construction, the construction method can provide reference and experience for the future similar projects.

**Keywords:** shield tunnel; city overpass; piles underpinning; site monitoring; settlement control

## 1. Introduction

With the rapid development of global urbanization, more and more people are migrating from the countryside to the central cities, especially in emerging developing countries. However, the rapid growing population puts great pressure on urban traffic system, especially in those cities and gradually becomes a bottleneck hindering the development of urban construction. In order to alleviate traffic pressure, many central cities began to develop their own rail transit system especially the metro tunnel. However, due to the limitation of city construction land resource, the metro tunnel route planning and design are becoming more complicated than ever. For instance, the new planning metro line often needs to cross through the old city district which already has a large number of existing structures, such as bridges, railways, roadways, buildings, etc. Those problems exist in almost all the new emerging cities all over the world and are huge challenges to the urban subway transportation development.

Generally speaking, the complex construction environment and conditions are the biggest challenges in metro construction especially when it is required to cross through the foundation pile of existing buildings or bridges [1–4]. The previous researches mainly focus on the theoretical empirical formula [5–8], laboratory similarity experiment [9,10], numerical simulation [11,12], and so on during the shield tunnel construction. In [13] presented a case of shield tunnel crossing through group piles foundation of a road bridge

by using numerical simulation and in-situ measurement in Shanghai. In [14] studied the tunnel-pile interaction considering the effect of tunnel location on the tunnel-pile interaction. In [15] studied two different protective schemes for shield tunneling adjacent to overpass group piles in Tianjin Metro.

Even though there are many, investigators and researchers have conducted to the settlement studies and technical controlling methods for the existing overpass bridge piles foundation underpinning construction during shield tunnel construction, however the relevant cases and references are still lacking. In this study, a simple but effective construction scheme to solve the problem of reducing the impact of shield tunnel construction on the existing bridge structure the metro tunnel has been proposed. This work considers a real metro shield tunnel construction project located in Chengdu (China) as an example to study the influence of shield tunnel on the existing bridge piles foundation settlement. Different from the former researches [16,17], in this project, to preventing underpinning bearing platform from cracking, several prestressed steel strands are set inside of it. In addition, a finite element simulation model which takes the whole construction steps, geological conditions, shield excavating methods, prestressed steel loads and overpass upper loads into consideration has been established to predict the overpass piles foundation settlement and take effective safety measures for the projects related to shield tunnel crossed foundation piles. As the numerical simulation model shows, the overpass piles foundation underpinning plan during shield tunnel crossing through are safe and feasible. Furthermore, the actual field settlement monitoring data on the existing overpass structure from piles foundation underpinning of shield tunnel crossing through the piles area is shown. Eventually, it is found that the field settlement measurement data which are taken from the existing bridge structure are highly consistent with the numerical simulation results. The numerical simulation analysis methods and field settlement controlling measures in this paper can indeed provide references and experience for the future similar engineering construction projects.

## 2. Project Overview

### 2.1. Engineering Background

The background subway engineering project is a round section metro tunnel that adopted shield construction method. The subway tunnel shield segments are prefabricated with C50 high-strength waterproof concrete which has an outer diameter 830 cm, inner diameter 750 cm and thickness 40 cm. The minimum buried depth of the subway tunnel is 11.8 m. Due to the urban route planning reasons, the subway tunnel must cross through the existing overpass piles area. To avoid overpass bridge structure being damaged, some critical measures which could reduce the impact on the existing structure must be taken. The existing bridge structure is part of the city interchange overpass ramp that has a one-way two-line design plan. As shown in Figure 1, the bridge is a simply supported box girder structure, and the superstructure is 1200 cm tall and 850 cm wide. The bridge structure includes a dimension of 6.5 m $\times$ 2.5 m $\times$ 2 m bearing platform supported by two drilled grouting piles which is 18 m long and 1.5 m in diameter placed in both sides. Due to the route planning design, the shield tunnel must cross through the foundation piles area and the original piles will be cut off. In order to reduce the impact of shield tunnel construction on the original bridge structure, a plie foundation underpinning design plan is carried out. Bases on the design plan, a new 21.0 m $\times$ 3.7 m $\times$ 3.0 m bearing platform is built to replace the existing one. Underneath the new bearing platform, two supporting piles which has a diameter of 2.0 m and a length of 16.0 m are built as the new structural foundation piles which were connected by the joint caps at the both sides.

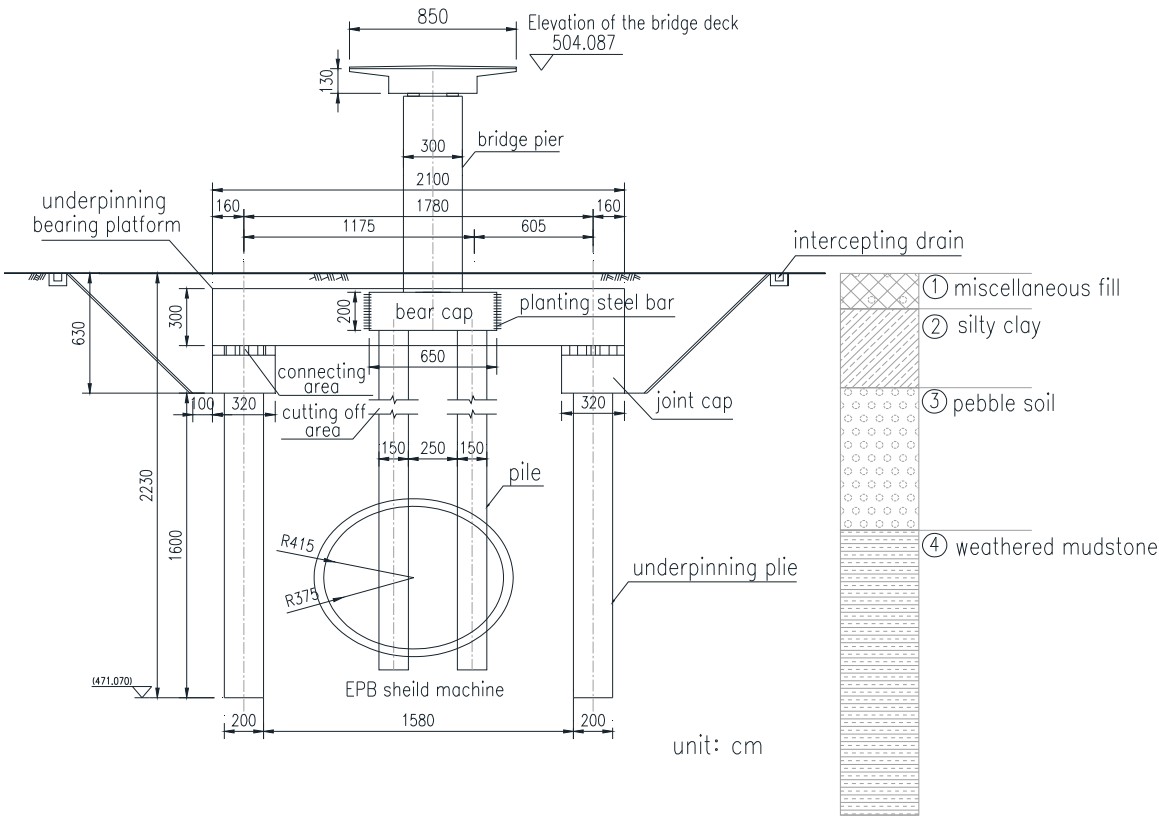

**Figure 1.** Elevation view of tunnel crossing viaduct foundation piles.

### 2.2. Engineering Geological Conditions

As the engineering geological investigation report shows, the stratum from ground surface to the depth of 28.4 m is divided into 4 main layers in terms of soil characteristics: ① miscellaneous backfilling layer containing a small amount of construction waste, stone debris and other impurities ranging from 1.0 m~2.3 m in thickness; ② silty clay layer with a few gravel fragments ranging from 2.85 m~4.2 m; ③ cobblestone soil layer with an average thickness of 7.5 m ranging from 4.8 m~12.3 m; ④ medium to slightly weathered rock layer. According to the site geological survey results, the rock layer is relatively hard and presented blocky structure. There is no weak interlayer, void or fractures found in the rock layer. And the 4th rock layer is the bearing layer for the original bridge piles foundation in this construction project. The main soil mechanical properties are listed in Table 1.

**Table 1.** Mechanical parameters of soil strata.

| Soil Name | Unit Weight (kN/m³) | Cohesion (kPa) | Friction (°) | Compressive Modulus (MPa) |
| --- | --- | --- | --- | --- |
| Miscellaneous fill | 16.5 | 10.5 | 22.2 | 5.0 |
| silty clay | 18.8 | 15.6 | 18.9 | 15.6 |
| cobblestone soil | 20.6 | 60.0 | 25.0 | 52.6 |
| Weathered rock layer | 21.4 | 150.0 | 30.2 | 180.0 |

### 2.3. Existing Problems and Solutions

#### 2.3.1. Existing Problem

The difficulty of shield tunnel construction in this project is the original bridge foundation piles. Generally, in the past, there are two preferred methods to solve this problem. The most economical and reasonable method was to modify the route line to avoid the piles obstacles. And the other solution is to demolish the original bridge structure when the

shield tunnel crossed through and rebuild the bridge after the shield tunnel construction was finished. However, both the traditional methods seem to be unacceptable because of the following reasons: (1) the construction shield tunnel is located in the central city due to the limited building land and resources in the urban area, which caused problem in metro route line planning, so it is impossible to modify the metro line to detour the original bridge foundation piles. (2) the traditional demolishing and rebuilding plan cost lots of money and time, therefore this method is also undesirable.

2.3.2. Solution

Taking all of the above factors into consideration, this paper proposes an acceptable and feasible piles foundation underpinning design plan to solve this problem. It makes fully consideration on the metro route line design, construction cost and the safety of the bridge structure. According to the underpinning and reinforcement plan, two new supported piles and a bearing platform should be built before the construction of the shield tunnel, after those new foundation structures were finished, connected the new bearing platform with the original one, and used small hydraulic equipment to cut the original bridge piles off, and completed the bridge structure system stress-conversion. In order to minimize the impact of construction on the bridge structure, a series of monitoring devices are set up and conducted, including settlement and displacement detectors on the pile cap and bridge pier, concrete inner stress of the rebuilding bearing platform and stress changes on the original bridge pier. Bases on the Code for Monitoring Measurement of Urban Rail Transit Engineering GB50911-2013 (Chinese national standard) the allowable settlement value of bridge pile cap is ±2 mm/d and the cumulative value is ±15 mm. In this project, the settlement and displacement control are the primary factors during the shield tunnel constructed, and the specific scheme is described as follow.

**3. Construction Scheme of Piles Foundation Underpinning**

In order to ensure the safety of the original bridge structure during the shield tunnel construction, it is eventually decided to perform a pile truncation and replacement scheme on the existing bridge piles foundation underpinning construction. The original bridge piles structure will be cut off and replaced by a new bearing platform including two piles before the shield tunnel crossed through. The main construction processes are described as follows:

(1)    Foundation excavation

To meet the construction space requirements, it is necessary to excavate and create an operation area before the pile truncation and underpinning construction. According to the construction design plan (Shown in Figure 1), the excavation slope of the foundation pit is 1:1 and the plane size is 32 m × 18 m. Before the pit excavation, the surrounding intercepting and barrel drain will be required to prevent foundation pit from being affected by the surface water.

(2)    Underpinning pile construction

In the urban area, due to the environmental pollution, it is essential to reduce mud discharge during piles construction. Thus, in this project, rotating drill is used to excavate underpinning piles on both sides of the underpinning pile cap. The underpinning piles are 2.0 m in diameter and 16.0 m in length.

(3)    Underpinning pile cap construction

Reinforce steel bars are planted into the original bearing platform to ensure the underpinning bearing platform to connect with the existing one. A total of 10 sets of 21–15.2 prestressed steel strands are placed at the bottom of the underpinning bearing platform to minimize the tensile stress on top of the new bearing platform. The prestress steel strands are arranged in two layers, and the design control stress is 1334.0 MPa. After the underpinning cap concrete reached the design strength, tensioned prestressed

steel strands, the underpinning bearing platform design and prestressed steel strands arrangement are presented in Figure 2.

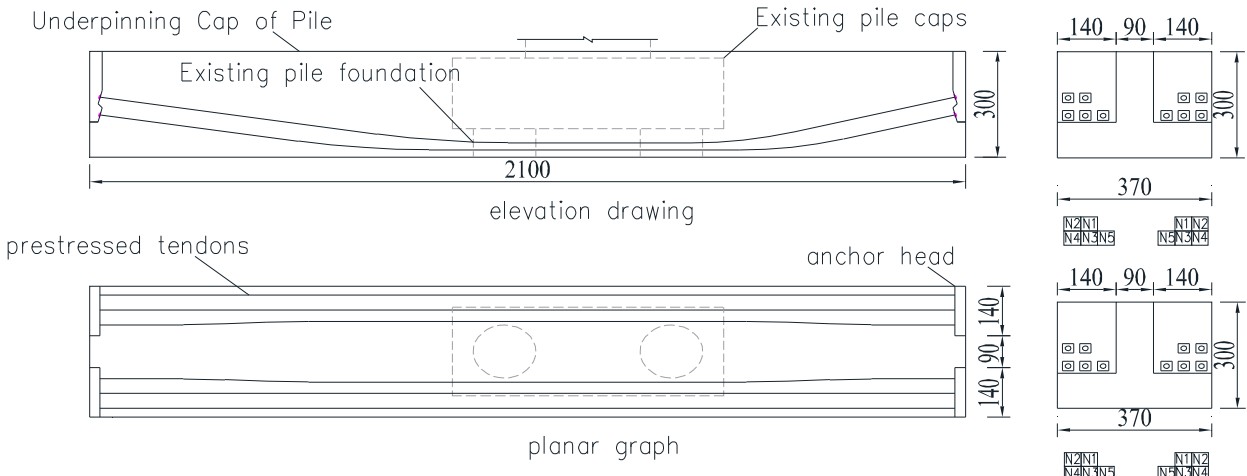

**Figure 2.** Design plan of the underpinning bearing platform.

(4)    Underpinning foundation pile cap pre-jacking construction

After the prestressed tension operation is completed, the most important step will be the underpinning foundation pre-jacking construction. In this project, the hydraulic jack for piles foundation underpinning construction design pre-jacking force $f_0$ = 1102 kN which is dominated by the bridge superstructure loads. The initial pre-jacking operation is carried out in a hierarchical loading method.

Based on the pre-jacking construction design, the initial supporting force of the new bearing platform pre-jacking operation is 10% of the design pre-jacking force and the pre-jacking supporting force is applied 10% $f_0$→30% $f_0$→60% $f_0$ step by step. While the new bearing platform is lifted up, the displacements of the bearing platform and the stress changes of the existing bridge pier should be monitored. When the initial pre-jacking operation is finished, all the bridge structure and the safety of the pre-jacking system should be checked. And then, the final pre-jacking operation could begin.

The initial supporting force of final pre-jacking operation is 60% $f_0$ and loaded by 10% $f_0$ each time till the jacks reaches the design supporting force. In the final pre-jacking operation process, each loading duration is at least 15 min and the displacement of the pile cap should be limited within 1 mm. When pre-jacking force reaches the designed force and the bearing platform is stabilized to lock the whole jacking system.

(5)    The existing bridge foundation pile truncation

While the underpinning piles foundation pre-jacking constructions is completed, the original foundation will be truncated and removed. Generally, small vibration cutting machine or static crush method can be used in truncating piles, however due to the limited time, the former method is chosen in this project.

Meanwhile in site displacements and stress changes monitoring will be required while the bridge piles are Truncating. According to the site monitoring data if there is an abnormal change in displacements and stress test results, the piles truncating operation must be terminated immediately. The piles truncating operation would not be proceed until the cause is identified and the mistakes are corrected. The main steps to construct the bridge foundation underpinning are shown in Figure 3.

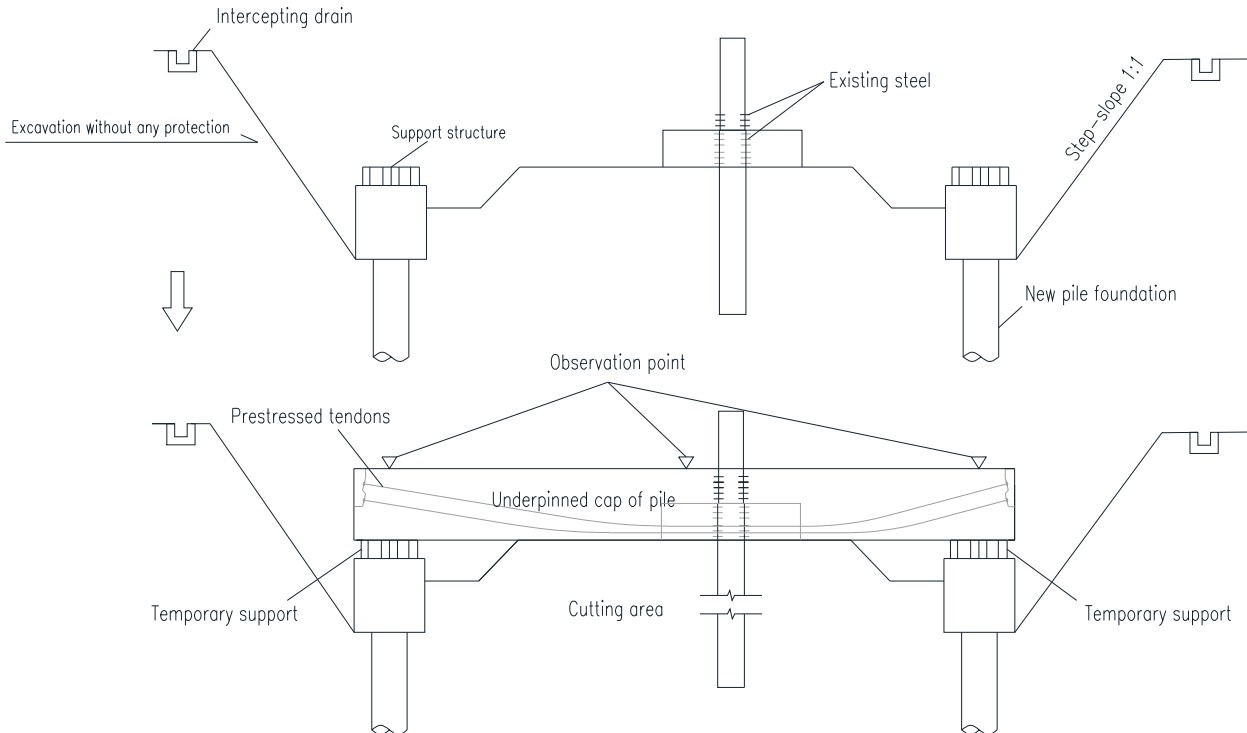

**Figure 3.** Construction process of the pile foundation underpinning.

(6) Piles sealing and backfilling construction

Before the bridge piles foundation underpinning construction was completed, the last step is sealing piles and backfilling.

In this process, the new underpinning foundation bearing platform would be connected with the new pile cap by pouring concrete into the connecting area under the new building bearing platform both sides. When the force given by the connecting area concrete reaches the designed strength, the supporting jacks can be slowly unloaded. Then the excavating area should be backfilled and the bridge foundation construction can be completed.

(7) Shield tunnel construction

Shield tunnel that crosses through the existing bridge piles foundation could be started after the sealing piles and pit backfilling process. Checking the whole bridge foundation structure underpinning construction qualities comprehensively before shield machine approached. Field monitoring data from the bearing cap and bridge pile such as settlements, displacements and stress changes are critically vital. According to the design requirement if there is a settlement that over 2 mm/day or the cumulative value is over ±15 mm, the shield tunneling construction would be ceased until find the causing reason and make it corrected. The briefly schematic diagram of the piles foundation underpinning construction is illustrated in Figure 4.

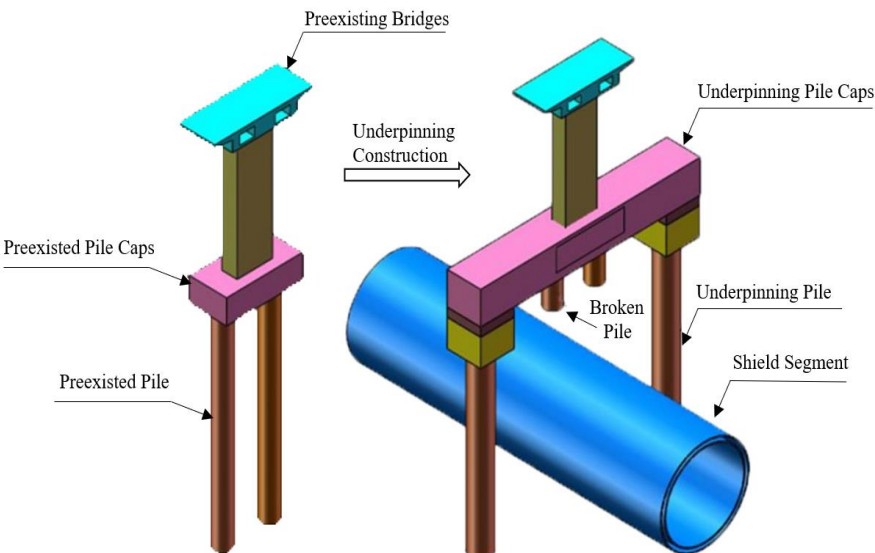

**Figure 4.** Schematic diagram of the pile foundation underpinning construction.

## 4. Numerical Calculation and Simulation

### 4.1. Finite Element Model

Bases on the actual situation of the shield tunnel crossing through overpass bridge piles foundation underpinning construction, a finite element calculation model is established used by Midas GTS.

A perspective view of the numerical calculating model is shown in Figure 5. The calculating model took soil mass, bridge deck, piers, bearing platform, foundation piles, shield tunnel, prestressed steel strands, excavating area, groundwater and construction steps into consideration. The dimension of this model is 85 m × 45 m × 30 m (X, Y, Z direction), and the mesh applies in this numerical calculating model are consisted of 115,843 nodes and 130,881 elements. To simulate the boundary condition, the horizontal displacements of the four vertical boundaries and the bottom boundary are fixed, and the top boundary is free.

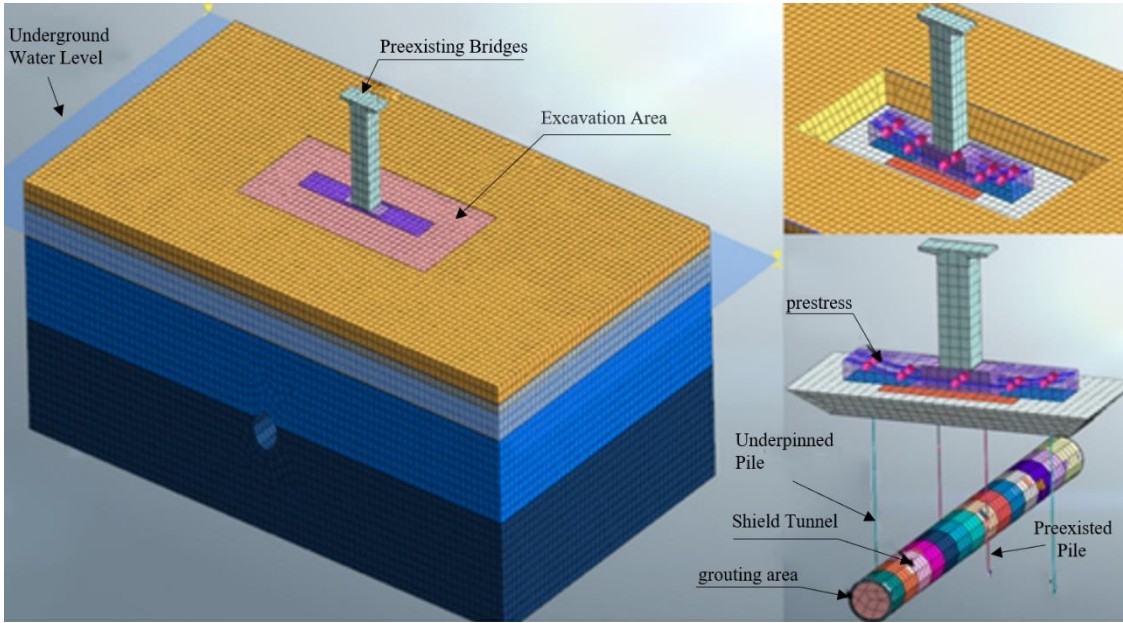

**Figure 5.** Numerical simulation model of pile foundation underpinning.

### 4.2. Material Properties

The material properties are defined according to the actual situation in the numerical model. There are four layers of rock and soil layer that were considered as the ideal elastoplastic material, and the Mohr-Coulomb yield criterion was used as its constitutive model based on the site geological survey data. The bridge structure including foundation piles, bridge pier and bearing platform are simulated as 3D block elements which are considered as linear isotropic elastic material. Meanwhile the prestressed steel strands that are placed at bottom of the underpinning bearing platform and shield tunnel including shield shells and segments are also presented in this simulation model. The main materials physical and mechanical properties are shown in Table 2.

**Table 2.** Physical and mechanical properties of materials.

| Name | Bulk Density $\gamma$ (kN/m$^3$) | Elastic Modulus $E$ (MPa) | Poisson Ratio ($v$) | Constitutive Relation |
|---|---|---|---|---|
| Existing bridge | 24.5 | $3.45 \times 10^4$ | 0.20 | Elastic |
| Underpinning pile cap | 24.5 | $3.0 \times 10^4$ | 0.20 | Elastic |
| Shield segments | 78.0 | $3.45 \times 10^5$ | 0.30 | Elastic |
| Shield shells | 78.0 | $3.45 \times 10^5$ | 0.30 | Elastic |
| Shield grouting layers | 24.0 | 30.0 | 0.20 | Elastic |
| Steel Strand | 78.0 | $1.95 \times 10^5$ | 0.30 | Elastic |

(Note: The mechanical parameters of rock and soil mass are defined with reference to Table 1).

### 4.3. Construction Stages

To study the influence of the original bridge structure settlements that causes by the shield excavation and foundation underpinning design plan, it is necessary to take the full construction steps into consideration by using numerical simulation model. Bases on the site actual construction scheme, it takes 15 construction stages to simulate the whole process of shield tunnel crossed through the existing bridges foundation areas and the mainly construction steps are described as follows: (0) The initial phase, to simulate the initial phase of the existing bridge and the displacement was wiped out to eliminate the impact that may be caused by the following construction steps. (1) Foundation excavation, to excavate the bridge foundation area and foundation underpinning working space can be created. (2) Underpinning piles and cap construction, to construct the underpinning piles and cap under the bearing platform on both sides. (3) Prestress tension construction, to tension the prestressed steel strands placed at the bottom of the underpinning bearing platform. (4) Bridge foundation piles underpinning and replacement, to start the jacks placed under the underpinning bearing platform and to load it to the designed force, then to complete the bridge foundation replacement process. (5) Existing foundation piles truncation, to passivate the original bridge foundation piles in simulation model. (6) Seal piles construction, to connect the underpinning bearing platform with the new pile caps on the both sides, and to unload the jacks. (7) Shield tunneling initial construction, to launch the shield machine and dug into the existing bridge foundation piles area. (8) Shield tunnel crossing through construction, to continue the shield construction and let it cross through the existing bridge foundation piles area. (9) Completed the construction of shield tunnel under the existing bridge pile foundation part. (10)~(14) Shield machine moved away from the existing bridge piles foundation area.

### 4.4. Calculating Load

(1) Self-weight

The self-weight coefficient of the calculation model is g = −9.81 m/s$^2$.

(2) Superstructure load

The superstructure load is the main factor that causes the settlement of the structure in construction period. In this project, the superstructure load of the bridge is converted into a surface load and applied to the top of the box girder. $f_s$ = 60.8 kN/m.

(3)　Steel strands prestress

The prestressed steel strands that places in the bottom of the underpinning bearing platform are 1860 PC steel wires. After the underpinning bearing platform concrete strength reaches the designed status, the steel wires are tensioned. The steel wires are tensioned by post-tensioning method and the design tension control value was $f_d$= 1334.0 MPa.

(4)　Pre-jacking force

During the piles foundation underpinning construction, the pre-jacking force control is the key point of the foundation replacement process. The pre-jacking force is loaded as a typical "step-by-step" way and the calculated pre-jacking force $f_p$ = 1102 kN.

(5)　Shield loads

The shield tunneling loads could be the primary factor that caused the bridge foundation structure deformed. Thus, it is essential to simulate the shield tunnel construction correctly. As shown in Figure 6, the simulation model took three kind of loads into consideration such as the tunnel face pressure, tunnel grouting pressure and the tunnel segments reaction force. Shield machine that uses in this project is earth pressure balance (EPB) shield, the outer diameter of the shield machine is 8.30 m, the inner diameter is 7.5 m, the width of the single segment is 1.6 m.

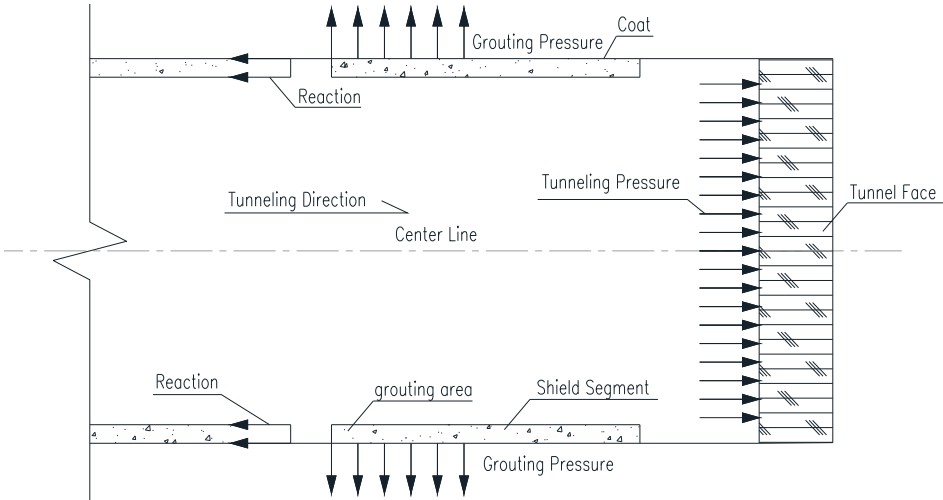

**Figure 6.** Sketch of shield excavation loads.

*4.5. Calculation Results*

(1)　Settlement calculation results

Figure 7 presents the calculation results of vertical settlements of the whole simulation model. It is shown that the settlement of the bridge causes by the tunnel construction starts from foundation excavation step and slowly increases till the piles foundation underpinning construction is completed. In the schemes of piles foundation underpinning plan, the maximum settlements of the bearing platform are about 5.73 mm and 5.71 mm.

In Figure 8 it presents the simulation model vertical settlements calculated results on the bridge pier point 1 and point 2. As it is shown that the bridge pier point 1 settlement increases from 3.51 mm to 4.71 mm after pier foundation underpinning constructions completed. Before the shield tunnel crosses through, the bridge pier settlement increases slowly. While the shield machine crossed through the foundation pile area, settlement on bridge pier obviously increases and reaches the maximum value, which is 5.28 mm. When compares with the settlements of point 2, they both show the same tendency.

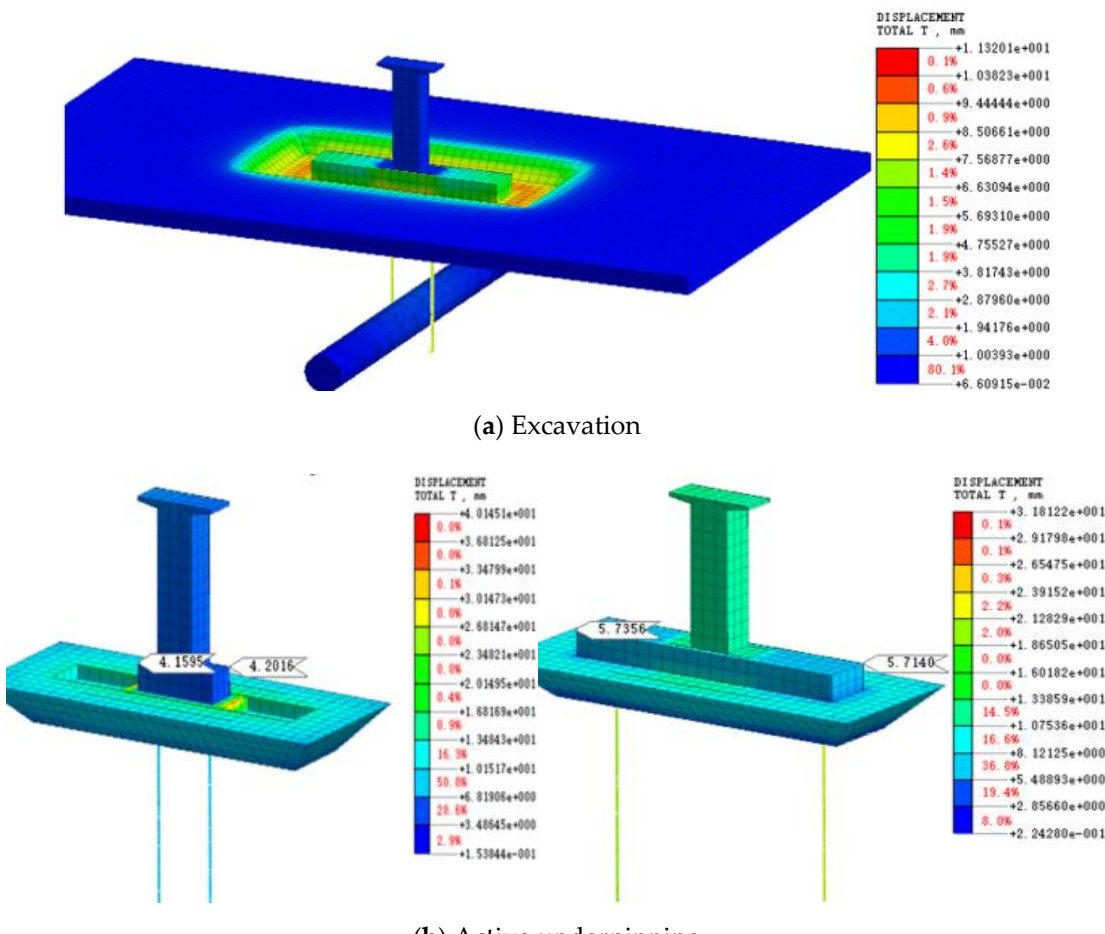

(**a**) Excavation

(**b**) Active underpinning

**Figure 7.** Vertical displacement of the numerical model.

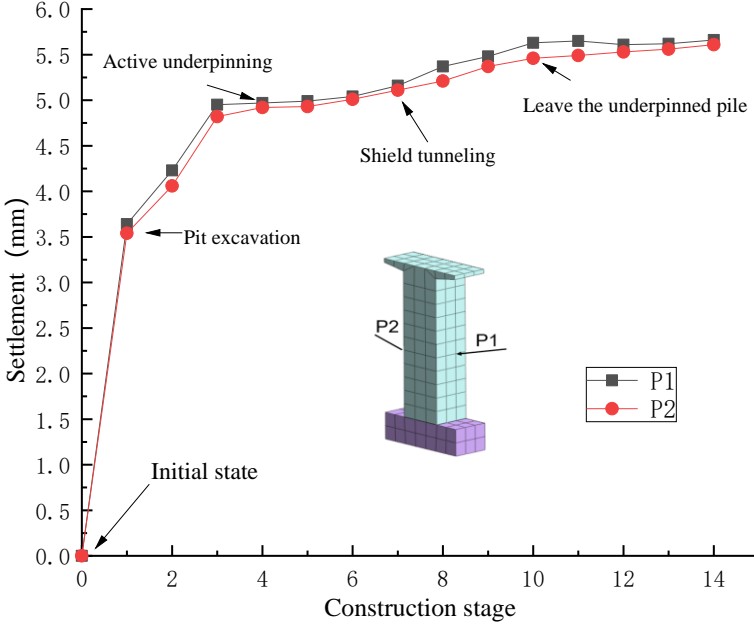

**Figure 8.** Vertical displacement of the existed pile cap.

In Figure 9, it presents the underpinning bearing platform settlements calculation results. As it shown the four measuring points settlement results were similar, the maximum settlements were about 6.81 mm.

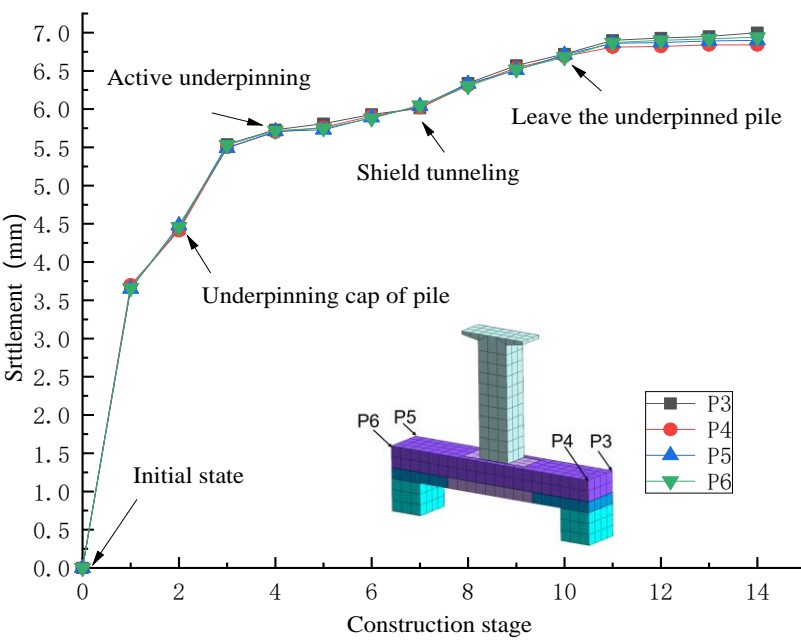

**Figure 9.** Vertical displacement of the underpinning pile cap.

(2)    Stress calculating results

As it presented in Figure 10, it shows the stress calculation result of the numerical model during the whole shield tunnel construction.

From the calculation result it can be seen that the underpinning platform concrete maximum tensile and compressive stress occurs at the foundation completed stage. After the prestressing system is established, the maximum tensile stress of the Sections 1-1 and 2-2 in the calculation model sharply reduces from the highest 1.12 MPa to the lowest 0.184 MPa due to the prestressing effect at the bottom of the bearing platform. Obviously, the prestressing system is conducive to control the bearing platform structure stress and prevent the underpinning bearing platform from cracking destruction under the superstructure loads. By comparing the bearing platform tensile and compressive stress calculation results that shown in Figure 10a,b the stress changing tendency are highly similar. The calculated maximal principle compressive stress of the bearing platform in Sections 3-3 and 4-4 were reduced from −6.89 MPa to −5.10 MPa.

Furthermore, the calculated results also shows that the shield tunnel construction will not cause significantly stress variations of the underpinning bearing platform. While the shield tunnel crosses through the original bridge piles foundation area, the bearing platform stress varies smoothly and the calculated maximal tensile and compressive stresses are 0.32 MPa and −5.71 MPa. Based on the C40 concrete design axial tensile stress $f_t$ = 1.79 MPa and design axial compressive stress $f_c$ = 19.1 MP, the bridge piles foundation underpinning design plan seemed to be reasonable and feasible.

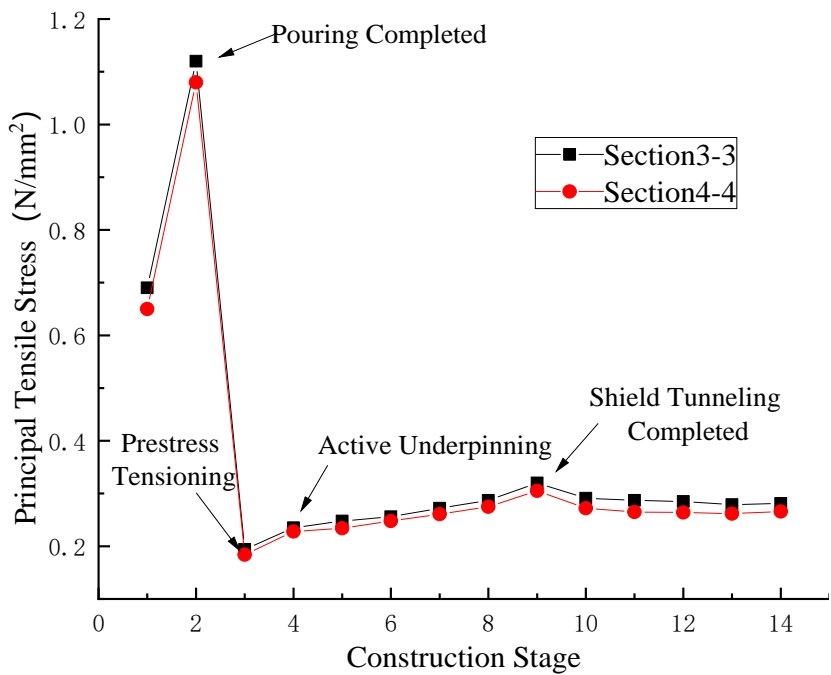

(**a**) underpinning bearing platform compressive stress (N/mm²)

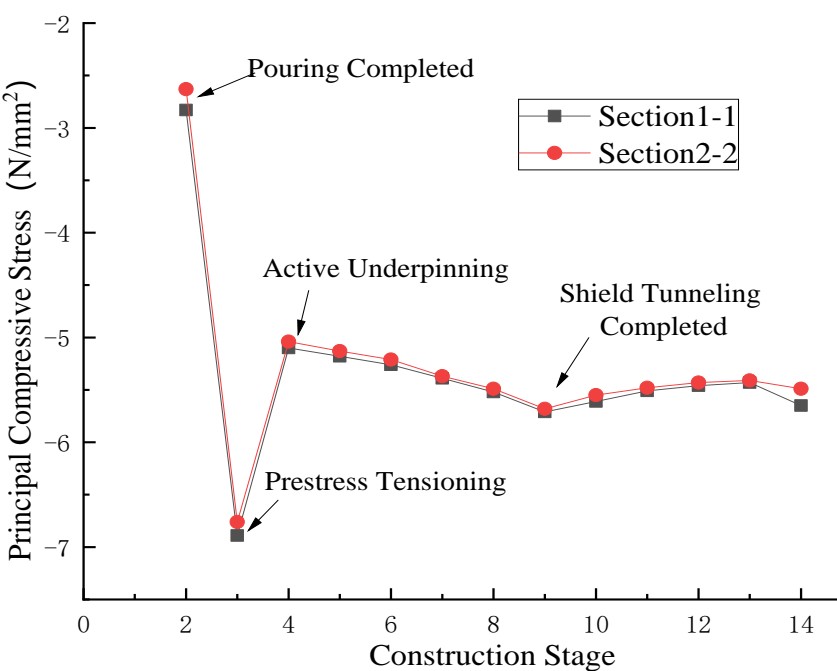

(**b**) underpinning bearing platform tensile stress (N/mm²)

**Figure 10.** Stress result of the numerical model.

## 5. Field Monitoring and Data Analysis

### 5.1. Site Monitoring Scheme

The underground shield tunnel crosses through the original bridge piles foundation area, due to safety reasons it is necessary to monitoring the structure internal state parameters such as vertical settlement on the underpinning bearing platform and stress changes of the underpinning concrete and the bridge pier structure during the shield tunnel excavation.

As it in Figure 11 presented, the layout of monitoring points for bridge pier and underpinning bearing platform. To investigate the effects of shield tunneling under the bridge, in this case, 6 settlement gauging points and 12 stress gauging points are set up.

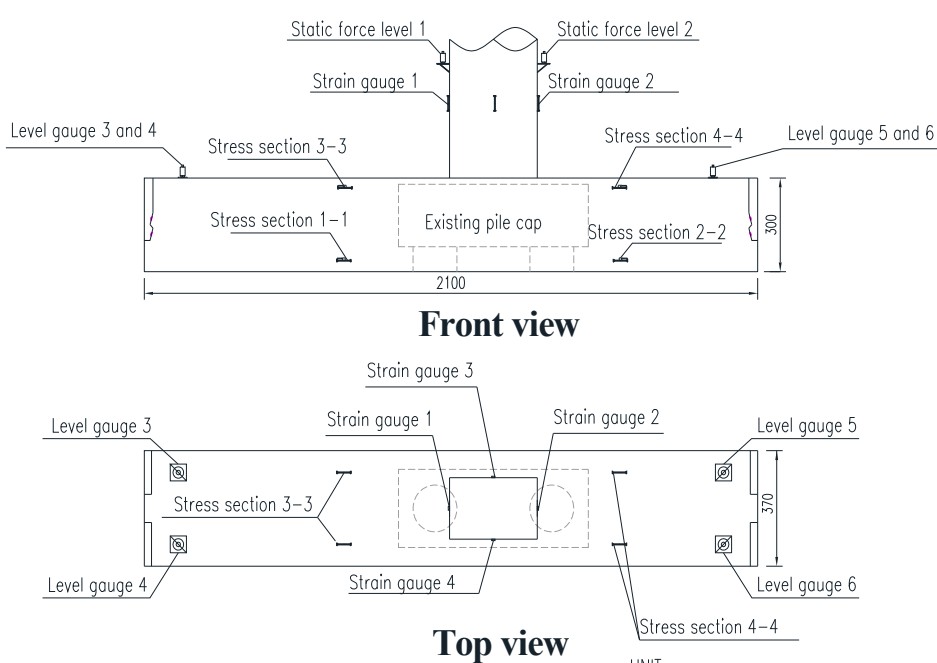

**Figure 11.** Monitoring sensor arrangement of the underpinning pile cap.

Settlement sensors no.1 and no.2 are placed on the bridge pier and sensors no.3 to no.6 are placed around the underpinning bearing platform to test the structure settlement changes during the shield tunneling. Due to the complexity of this foundation underpinning construction a new settlements collection system named intelligent hydrostatic level are adopted in this project to monitoring the settlement changes on the underpinning bearing platform and the pier 24 h.

Stress sensors no.1 to no.4 are set up around the bridge pier to observe the variation of stresses especially in the foundation actively underpinning and replacement stages. The other stress sensors are placed inside the underpinning bearing platform named Sections 1–4 which can track the stress changes during the whole construction.

*5.2. Data Analysis*

(1)  Settlement test

In bridge piles foundation underpinning construction site monitoring, an intelligent settlement monitoring system named hydrostatic level sensor JMDL-62XXAT is applied to measuring the settlement variation during shield tunneling construction. The accuracy of hydrostatic level sensor settlement measuring system is 0.01 mm and the effective measuring range is 50 mm, and it meets the shield tunnel construction requirements of this project.

As is illustrated in Figures 12 and 13, field monitoring settlements of pier point 2 and underpinning bearing platform point 5 varied from 3.71 mm to 6.48 mm while the shield machine was crossing through, and the range of subsidence on the bearing platform is 3.01 mm to 7.40 mm.

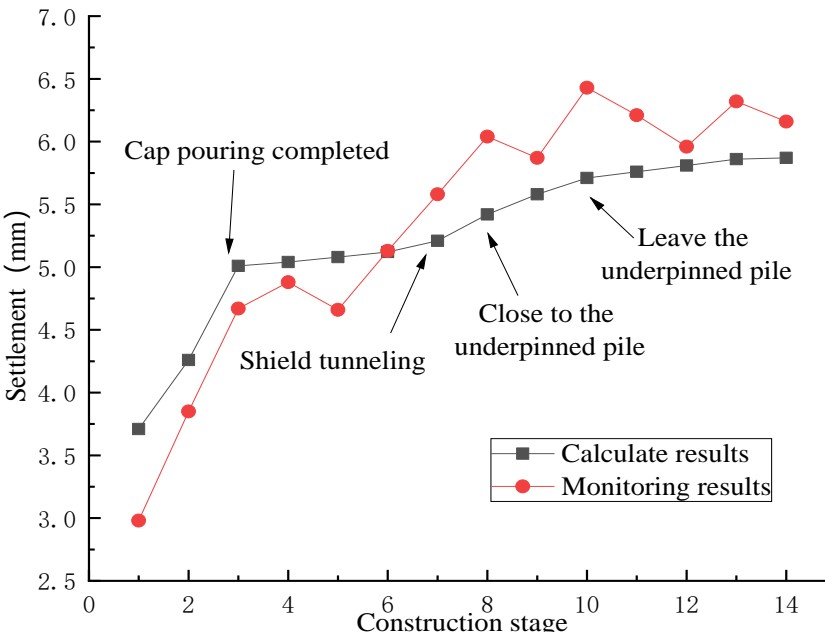

**Figure 12.** Vertical displacement results of the overpass pier.

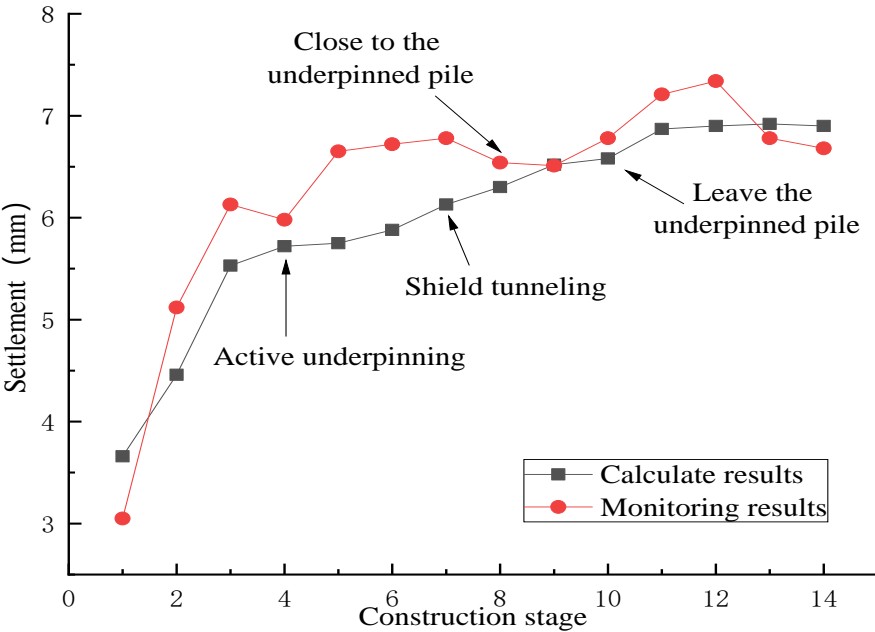

**Figure 13.** Vertical displacement results of the underpinning cap.

After comparison, the numerical simulation model's calculated results are consistent with the field monitored settlement date. This proves the reasonability and safety of the underpinning bearing platform construction scheme.

(2)    Stress test

In order to verify the safety of origin bridge structure during shield tunnel excavating, field stress tests are also necessary during the shield tunnel excavating. In this project, stress monitoring sensors named JMZX-212AT are adopted to gauging the stress changes of the bridge pier and the underpinning bearing platform. The stress sensor is suitable for measuring the surface and internal stress of concrete structure. The accuracy of the stress sensor was $1\ \mu \cdot \varepsilon$ and has an effective measuring range is $\pm 1500\ \mu \cdot \varepsilon$, which is also suitable for the requirement of this project.

Bridge pier stress monitoring started from bridge foundation excavation till the shield tunnel construction completed and the stress variations are gauged by four sensors numbered 1 to 4 that placed on the bridge pier surface. As it presented in Figure 14, the bridge pier stress test results fluctuated up and down near the zero point. The stress range of fluctuation is 0.2 MPa~0.3 MPa. The above phenomenon illustrates that the bridge pier is stable during the bridge piles foundation underpinning construction, and the shield tunnel excavation can barely affect the origin bridge structure. The bridge pier stress fluctuating phenomenon might be caused by stress sensors themselves, ambient temperature variation or external environment vibration and it is inevitable.

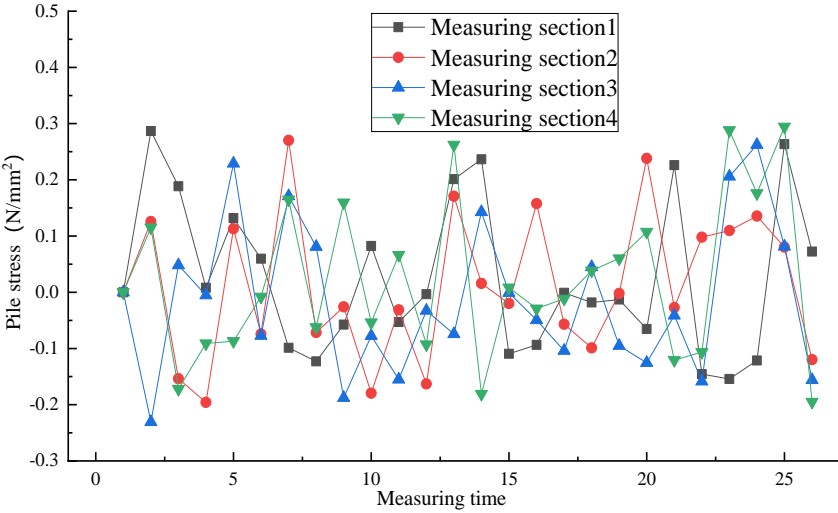

**Figure 14.** Stress gauging result of piles foundation.

In addition to test stress variation on the bridge pier, the underpinning bearing platform stress changes should also be monitored correspondingly to ensure the safety during shield tunnel construction. In this case, underpinning bearing platform stress sensors are placed in the top and bottom of the both sides numbered Sections 1-1–4-4.

The underpinning bearing platform stress monitoring data from Sections 1-1 and 2-2 are presented in Figure 15. As the stress curves shows, stress on the top of the underpinning bearing platform reaches the maximum before the prestress system was established and the maximal principal stress is 1.41 MPa. When the steel strands are prestressed, underpinning bearing platform principal stress decreases rapidly due to the influence of prestresses. And then, due to the relaxation of prestress and other reasons the stress of both sections gradually increases and reaches 0.38 MPa. However, during the shield tunnel excavating there is no abnormal stress changes. After the shield machine is away from bridge piles foundation area, stress test data is stable. And the field stress monitoring date is similar with the numerical calculate results.

Stress monitoring data of sensors numbered Sections 3-3 and 4-4 which places in the bottom of underpinning bearing platform are presented in Figure 16. As it shown that before the steel strands are prestressed, stress in the bottom of underpinning bearing platform reaches the maximum −8.02 MPa. While the prestressed system is completed Sections 3-3 and 4-4 stress decreases from −8.02 MPa to −6.20 MPa immediately. Furthermore, while the shield machine crosses through the foundation area there is no abnormal stress changes could be founded based on the field stress gauging data, stress of underpinning bearing platform is stable.

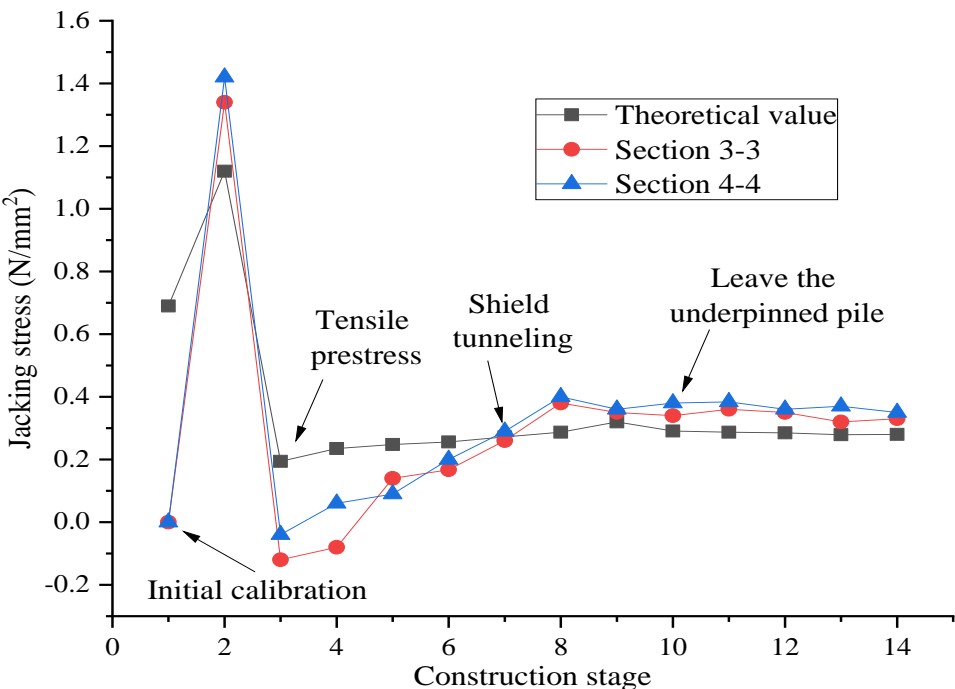

**Figure 15.** Tensile stress gauging results of piles foundation underpinning.

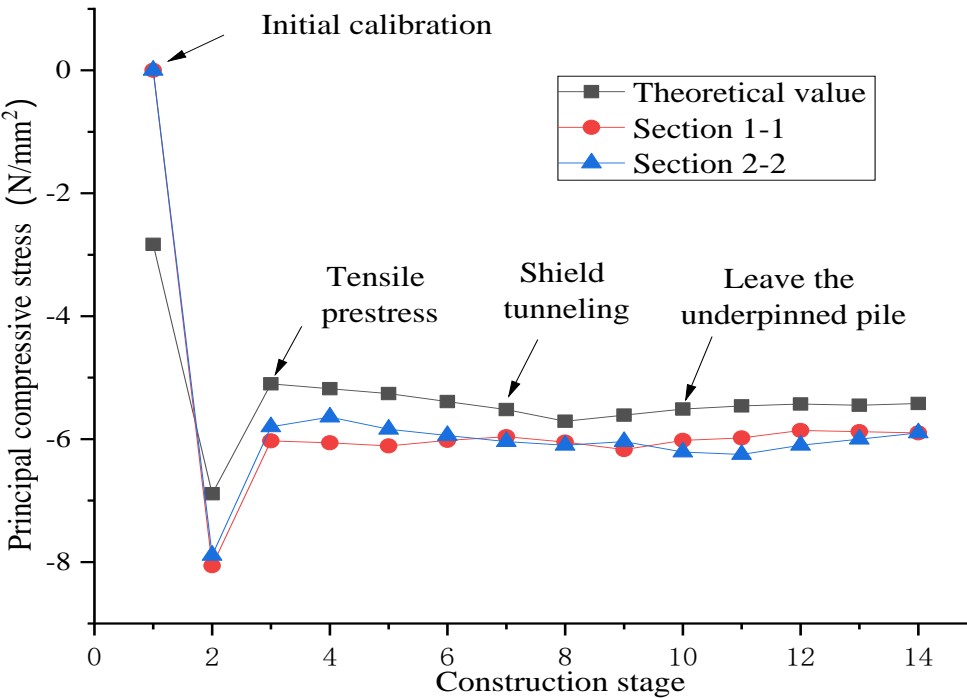

**Figure 16.** Compressive stress gauging results of piles foundation underpinning.

Comparison of monitoring data of four sections clearly indicates that the piles foundation underpinning scheme could effectively reduce the impact of shield tunneling on the original bridge structure and stress system could obviously improve the stress state of the underpinning bearing platform which could avoid structure being cracked or deformed and so on.

## 6. Conclusions

In terms of shield tunnel crossed through the existing bridge piles foundation area in this project a feasible and safety foundation underpinning scheme is carried out, according to the field monitoring data and numerical simulation results, the following conclusions are drawn:

1.  A numerical simulation model is established which took all the influencing factors into consideration, according to the site monitoring data from underpinning construction process that proves the feasibility and reasonability of the pile foundation underpinning construction scheme.
2.  To ensure the quality and safety of pile foundation underpinning construction work, a key step is to reducing the disturbance on the existing bridge structure during transfer the superstructure load of the existing pile foundation to the new underpinning foundation. Thus, choose a reasonable pre-jacking method is necessary. In this project, the load increment is added in stages which gives project construction engineers enough time to check the structural state of existing bridge and ensure the implementation of pre-jacking process.
3.  According to the site monitoring data, the settlements of the bridge structure are dominated by the load transfer process while the prestress system is established and shield machine disturbance during shield tunneling. The settlement caused by the pile foundation underpinning construction accounts for nearly half of the total settlement that indicates the importance of taking reliable pile foundation underpinning design plan.

As a conclusion, the maximal settlement in site monitoring of bridge structure during shield tunneling is 6.55 mm; the maximal concrete principal compressive stress and maximum principal tension stress of the underpinning bearing platform are 6.1 MPa and 0.36 MPa, respectively. Those results meet the settlement control target and concrete strength design index. Such methods could provide good reference and guidance for the construction of similar engineering in the future.

## 7. Discussion

The field monitoring stress date of underpinning bearing platform results indicates that the underpinning bearing platform is affected by the prestress system. However, there is an interesting phenomenon that after the establishment of prestressing system, the underpinning bearing platform settlement did not decrease 5~6 mm as calculated in the project design process, because of the effect that prestress would cause the middle of bearing platform arch up. On the contrary, the settlement of middle of underpinning bearing platform is just 1.0 mm bigger than support part and it is less more than design scheme expected. This phenomenon may be due to the differences of concrete materials quality of existing and new underpinning bearing platform. And this problem needs more test data and engineering practice to solve.

**Author Contributions:** Conceptualization, C.Z. and B.Z.; methodology, C.Z.; software, Y.Z.; validation, C.Z., B.Z. and Z.Z.; formal analysis, Y.Z.; investigation, Z.Z.; resources, B.Z.; data curation, Y.Z.; writing—original draft preparation, C.Z.; writing—review and editing, Z.Z.; visualization, Y.Z.; supervision, B.Z.; project administration, C.Z.; funding acquisition, B.Z. All authors have read and agreed to the published version of the manuscript.

**Funding:** This research was funded by the National Natural Science Foundation of China, grant number U1834207 and the Sichuan Province Youth Science and Technology Innovation Team Program, grant number 2017JY003.

**Acknowledgments:** The authors gratefully acknowledge the support of the National Natural Science Foundation of China (U1834207) and the Sichuan Province Youth Science and Technology Innovation Team Program (2017JY003). And the authors are grateful to the reviewers and editors for their valuable comments and suggestions that helped to improve the quality of the paper.

**Conflicts of Interest:** The authors declare that they have no conflict of interest or personal relationships that could have appeared to influence the work reported in this paper.

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
