# Peer review of "Case Study of Underground Shield Tunnels in Interchange Piles Foundation Underpinning Construction"

_applsci, doi:10.3390/app11041611_

Round 1
Reviewer 1 Report
- It is not clear weather the research approach is developed by the authors or not? If it is a novel approach then then the intellectual input should be specifically describe and elaborated with different findings. Otherwise, the contribution of the authors will not be stablished if the design is done by following conventional specifications. In simple language, what is the originality of the work?
- The motivation of the research appears the justification of the numerical model with the field data for future reference. Then the question is: Was not any numerical model developed during design and analysis of that practical project? Why the authors further need to justify the numerical model?
- In load calculation section 4.4, why superstructure load is not considered as dead load?
- The results are given as fact, no discussion and impact of any deviation is given. The consistency between the real data and modelling is well stablished but any scientific interpretation behind the trend of results are not well developed.
Author Response
Authors’ reply for Expert reviewer’s report
Dear Expert reviewer,
We have received your report about this article, thanks for your precious time and opinions. According to your report, the authors want to make following clarification and explanation.
NO1.
>Expert opinions of reviewers:
>It is not clear whether the research approach is developed by the authors or not? If it is a novel approach then then the intellectual input should be specifically described and elaborated with different findings. Otherwise, the contribution of the authors will not be stablished if the design is done by following conventional specifications. In simple language, what is the originality of the work?
>The authors reply:
Dear Expert reviewer:
This paper takes a subway project under construction as an example and presents a complex process of shield tunnel crossing through the existing bridge foundation piles area. The originalities of this work are reflected in the following:
- Different from other similar metro construction project (added references for example), in this project the authors’ team extra added steel pre-stressed strand to decrease the concrete tensile stress of underpinning bearing platform. And this method can prevent underpinning bearing platform from cracking and improve the appearance quality of construction project that is an important index in Chinese standard for quality assessment of engineering construction.
- In traditional engineering project, the most common settlements measurement of structure is adopted level. And this settlements measurement method needs professional surveyor conduct the operation that needs a plenty of time and not efficient. Thus, in this paper, the authors’ team used a new level measurement system named static force level which can automatically gauge settlement of structure without people completed the structure settlement measurement work and this could be an improvement in measurement method.
In the end, shield tunnel machine crossing through the bridge foundation piles area is a tough work, for the safety reason, both project engineers and managers are tending to choose more mature and reliable method to complete this work. Thus, the originalities of this work are more reflected in improvement of design plan and operate details. We hope that you could unstrand.
NO2.
>Expert opinions of reviewers:
>The motivation of the research appears the justification of the numerical model with the field data for future reference. Then the question is: Was not any numerical model developed during design and analysis of that practical project? Why the authors further need to justify the numerical model?
>The authors reply
> Dear Expert reviewer:
In China's construction project area, the project design plan that including route planning and engineering structural entity are provided by design institute, but the constructed methods are not including. Thus, the project construction company needs to take their own method to complete the project construction.
In other language, in this project, the project design institute won’t provide specific methods for project construction company to finish the shield tunnel crossing through the original bridge foundation piles area. Thus, there is no former numerical model.
In this project, because of the difficulty and complexity, the project construction company needs to find more professional and experienced team to make reasonable and safety method to guarantee the safety while the shield tunnel is crossing the existing bridge pile foundation area. Thus, they find the author’s university (Southwest Jiaotong University ) to help them make reasonable method to finish this difficult work. And based on the site actual situation, the authors’ team provided the reasonable and specific construction (as described in this article) method and established the numerical model according to the construction plan. In simple language, the specific construction method and numerical model are provided by the authors’ team original, not from the design period.
We hope our clarification is understandable, because the above situation might be different from your country, and we sincerely hope that you could understand.
NO3.
>Expert opinions of reviewers:
> In load calculation section 4.4, why superstructure load is not considered as dead load?
>The authors reply:
Dear Expert reviewer:
>The author changed the original Dead weight to Self-weight.
>Explanation:
The former describes Dead weight is not accurate, and according to the article the author changed the original describe Dead weight to Self-weight to make it more comprehensible.
NO4.
>Expert opinions of reviewers:
> The results are given as fact, no discussion and impact of any deviation is given. The consistency between the real data and modelling is well stablished but any scientific interpretation behind the trend of results are not well developed.
>The authors reply:
> Expert opinions of reviewers:
Thanks for your advices, the authors update and rewrite the article conclusion part, given the key step to ensure construction quality and discussed some interesting phenomenon during construction, and presented the authors’ opinions about this construction work that could be the priority for the further similar construction quality control.
>In addition: according to the opinions of expert reviewers the authors added three more relevant references to provide more background information.
>In the end, all the revisions are colored in red, and we would like to express our most sincere thanks for your support and advices.
Sincerely yours
Dr. Chengran Zhang
2021.1.29

Reviewer 2 Report
Respected Authors
As an engineer with university background in Mining and other geotechnologies I have two kinds of reservations concerning your study.
Some minor issues are related to editorial merit, quality of figures and clarity of descriptions.
figure 1 - underpinning pile (not plie)
figure 3 - the pile foundation underpinning (not Piles)
4.3. You refer to 9 construction stages (1) ... (9) - figures 8,9,10,12,13,15 and 16 refer to 14 or 12 constrution stages. Are the stages somehow comparable? Please clarify.
figures 8,9,12 and 13 should have unified ranges of values on horizontal axis. Concerning vertical axis (settlements) I'd suggest using "values in inverse order" with starting point at "0" value. That would be helpful to compare measured and observed values.
A major problem is related to the achieved "deflection form" of supporting prestressed beam. It seems that settlement of piles supporting it are bigger then final settlement of supported structure (bridge pillar). In standard static scheme of a centrally loaded beam I'd expect larger settlement of the centre of the beam and consequently supported structure then measured displacement of points of support. It is very confusing, could be (theoretically) caused by a large prestressing force in the beam causing pre-elevation of its central part but it is not really probable in the real case.
I marked major revision just to give you time to restructure the paper to make it more readable and clear for a reader who was not involved in the project and explain (I hope that it is possible) your position concerning unexpected shape of the beam after loading.
Sincerely
Author Response
Authors’ reply for Expert reviewer’s report
Dear Expert reviewer,
We have received your report about this article, thanks for your precious time and opinions. According to your report, the authors want to make following clarification and explanation.
NO1.
>Expert opinions of reviewers:
>Some minor issues are related to editorial merit, quality of figures and clarity of descriptions.
figure 1 - underpinning pile (not plie);
figure 3 - the pile foundation underpinning (not Piles);
> The authors’ reply:
According to the expert reviewer, the authors’ modified the former mistakes.
4.3. You refer to 9 construction stages (1) ... (9) - figures 8,9,10,12,13,15 and 16 refer to 14 or 12 constrution stages. Are the stages somehow comparable? Please clarify.
>The authors’ reply:
Dear Expert reviewer,
In the former manuscript, the author describes the numerical model has 9 stages, that is because the author omits several stages of shield machine moves away from existing bridge. Accord to your advices, we realize that the above treatment method is not appropriate. Thus, the we modified the manuscript, and updated every construction stages’ description, and make it is more easier to understand for the other readers.
NO2.
>Expert opinions of reviewers:
>figures 8,9,12 and 13 should have unified ranges of values on horizontal axis. Concerning vertical axis (settlements) I'd suggest using "values in inverse order" with starting point at "0" value. That would be helpful to compare measured and observed values.
>The authors reply:
> Dear Expert reviewer,
According to your suggestion, the author changed some figures in this article and starting point at zero value.
NO3.
>Expert opinions of reviewers:
> A major problem is related to the achieved "deflection form" of supporting prestressed beam. It seems that settlement of piles supporting it are bigger then final settlement of supported structure (bridge pillar). In standard static scheme of a centrally loaded beam I'd expect larger settlement of the centre of the beam and consequently supported structure then measured displacement of points of support. It is very confusing, could be (theoretically) caused by a large prestressing force in the beam causing pre-elevation of its central part but it is not really probable in the real case.
>The authors reply:
> Dear Expert reviewer:
Your opinion about this unexpecting settlement is definitely right. In fact, during the project design phase, after the establishment of prestressed system the center of the underpinning bearing platform we speculated that the middle of underpinning bearing platform will arch up for 5~6mm.
However, the truth is the settlement of middle of underpinning bearing is just about 1mm than support area.
We assume the above phenomenon may cause by the differences of concrete materials quality of existing and new underpinning bearing platform. Unfortunately, we don’t have enough test date to confirm our hypothesis. However, you still provided us a very useful suggestion for such researches. We are truly hoping that we could do further researches as you pointed out.
And those concerns we added to the conclusion part as a discussion. Thanks again for your precious time and advices.
>In the end, all the revisions are colored in red, and we would like to express our most sincere thanks for your support and advices.
Sincerely yours
Dr. Chengran Zhang
2021.1.30

Round 2
Reviewer 2 Report
Dear Authors
It seems to me that you corrected most of the editorial issues. There is still one problem in Figure 1.: underpinning plie instead of underpinning pile.
I appreciate the standardization of Figures with charts concerning displacements. The x-axis is ok now - asyou refer to the same range of finally well-defined construction stages.
I'd still insist on some standardization of y-axis. I'd suggest 8 mm to make it comparable between charts. But it is just a gentle suggestion.
I'm glad that you also noticed the problematic difference between settlements of pile caps and settlements of the bridge pillar over the center of the loaded beam. In the current version of the paper it is somehow discussed and does not bring any more confusion.
Best regards